# Intensification of Dry Reforming of Methane on Membrane Catalyst: Confirmation and Development of the Hypothesis

**DOI:** 10.3390/membranes12020136

**Published:** 2022-01-23

**Authors:** Natalia Gavrilova, Sergey Gubin, Maria Myachina, Valentin Sapunov, Valery Skudin

**Affiliations:** 1Department of Colloid Chemistry, Faculty of Natural Science, D. Mendeleev University of Chemical Technology of Russia, Miusskaya sq., 9, 125047 Moscow, Russia; myachina.m.a@muctr.ru; 2Department of Chemical Technology of Carbon Materials, Faculty of Petroleum Chemistry and Polymers, D. Mendeleev University of Chemical Technology of Russia, Miusskaya sq., 9, 125047 Moscow, Russia; s.gubin@rambler.ru (S.G.); skudin.v.v@muctr.ru (V.S.); 3Department of Chemical Technology of Basic and Petrochemical Synthesis, Faculty of Petroleum Chemistry and Polymers, D. Mendeleev University of Chemical Technology of Russia, Miusskaya sq., 9, 125047 Moscow, Russia; sapunovvals@gmail.com

**Keywords:** membrane catalysis, membrane catalyst, thermal creep, Knudsen diffusion, dry reforming of methane, intensification

## Abstract

This article presents an analysis of kinetic studies of dry methane reforming (DRM) in a reactor with a membrane catalyst (RMC) in the modes of a contactor with “diffusion” and “forced” mass transfer. Comparison of the specific rate constants of the methane dissociation reaction in membrane and traditional reactors confirmed the phenomenon of intensification of dry methane reforming in a membrane catalyst (MC). It has been experimentally established that during DRM, a temperature gradient arises in the channels of the pore structure of the membrane catalyst, characterized by a decrease in temperature towards the inner volume of the MC, and initiates the phenomenon of thermal slip. The features of this phenomenon are highlighted and must be considered in the analysis of kinetic data. The main provisions of the hypothesis explaining the effect of intensification by the occurrence of thermal slip in the channels of the pore structure of the MC are formulated. The proposed hypothesis, based on thermal slip, explains the difference in rate constants of traditional and membrane catalysts, and substantiates the phenomenological scheme of DRM stages in a reactor with a membrane catalyst.

## 1. Introduction

The use of membranes and membrane catalysts (or catalytic membranes), which is always accompanied by the division of the reactor volume into two parts, creates conditions for controlling the transport of substances in the pore structure of such partitions (diaphragms) by changing the differences of pressures, concentrations, or temperatures in these adjacent parts. In studies of membrane catalysis, pressure change is more often used [1]. Changing the concentration difference on both sides of the membrane catalyst can be achieved by using a sweep (purge) gas [2,3,4]. The use of the temperature difference as a driving force for the transport of gases in the pore structure of the membrane is still extremely rare in the membrane apparatus [5].

The modern theory, explaining the advantages of membrane catalysis over the traditional one, suggests that the main mechanism of membrane influence in a reversible reaction is the selective transport from the reaction zone of one or several products of the mixture through the pore structure [6,7,8]. Removal of reaction products and rejection of the initial reagents in the reaction mixture change the chemical equilibrium, shifting it towards the direct reaction. It is generally accepted that the displacement of equilibrium concentrations when using a membrane depends on a phenomenological coefficient—the separation factor. At the same time, in membrane gas separation processes, the transport characteristics of membranes (ideal or real separation factors) are usually determined under isothermal conditions.

The separation factor reflects the ratio of the key components of the reaction mixture in the permeate and in the retentate in the membrane reactor. The actual separation factor is determined using real gas mixtures. Sometimes, when comparing membranes, a separation factor is used for a mixture of real neutral gases: for example, a mixture of N_2_ and H_2_, which are not directly related to the reaction carried out in a membrane reactor [9]. Such an assessment shows the relative effectiveness of membranes but is not always an objective assessment under the conditions of a chemical reaction.

It is assumed that the separation factor can be controlled by changing the pore characteristics of the membrane or membrane catalyst. However, the achievement of the maximum values of the separation factor is often accompanied by a decrease in the specific productivity of the membrane catalytic reactor, which makes its use economically unprofitable [10].

This hypothesis is widespread and primarily refers to membrane reactors in which the concept (mode) of the extractor is implemented. The use of the separation factor in the analysis of catalytic reactions in a membrane reactor does not seem entirely correct, since changes in the concentrations of the components of the reaction medium on both sides of the membrane catalyst occur simultaneously, both as a result of a chemical reaction and in mass transfer processes [2,9,11]. This approach is not practically applicable to other modes of membrane reactors such as contactor and distributor [3,12,13]. It is difficult to explain the intensification of the catalytic reaction in a reactor with a membrane catalyst in the contactor mode by a change in chemical equilibrium alone, when the initial mixture of reagents is forcibly transported through the pore structure of the membrane catalyst [14,15], and the resulting reaction mixture leaves the reaction volume without separation.

Many industrial reactions are high-temperature and subject to kinetic limitations due to the inhibition of diffusion in the pores of traditional catalysts. Diffusion limitations on the transfer of substances in the porous structure of the catalyst can allow reaching the equilibrium degree of methane conversion created by the membrane only at low values of the supplied mixture of reagents or long contact times. Considering chemical transformations in membrane reactors, many authors a priori assumed isothermal conditions in the porous structure of both membranes and membrane catalysts, even when studying reactions accompanied by large thermal (exo- or endo-) effects [9,16].

The choice of a criterion for assessing the intensification of catalytic processes when using membranes and membrane catalysts is of great importance for processes in reactors with membrane catalysts, which can operate in different modes, in contrast to a traditional reactor. The use, in many studies, of criteria for comparing the degree of conversion and selectivity, which are relative indicators, does not give a complete picture of the intensification of catalytic processes. The most objective estimates that need to complement the comparison of membrane and traditional catalytic reactors are indicators that characterize the rate of processes in the compared devices. This is the turnover frequency (TOF) and/or the specific rate constant of the target reaction or rate-limiting stages of the catalytic process carried out in the reference and comparison reactors. In the same way, different modes or concepts of membrane reactors can be compared. Application of kinetic experiment and analysis to membrane reactors provides more complete information about the intensification of processes, and in the general case, allows considering the mechanisms of chemical and mass transfer stages.

The hypothesis on the causes and mechanism of intensification of dry reforming of methane (DRM) in a reactor with a membrane catalyst (RMC) in the publication [17], was substantiated by a kinetic experiment in traditional and membrane reactors. Comparison of the rate constants of the rate-limiting reaction (methane dissociation) in the DRM process in membrane and conventional reactors was explained by the intensification of the mass transfer stages of the process, which was confirmed [18] by the appearance in the porous wall of the membrane catalyst (MC) of one of the types of non-isothermal Knudsen diffusion—the phenomenon of thermal slip (other names for this phenomenon are thermal transpiration or thermal creep).

When comparing membrane and traditional catalysts with the same active substance (tungsten carbide—WC), the identity of the chemical laws of the main reactions of the DRM process was established. It followed from the experiment that the rate constant of the initial rate of the methane dissociation reaction on the membrane catalyst turned out to be 30 times higher than that on the traditional (powder) catalyst of the same composition. Analysis of the characteristics of the pore structure of the membrane catalyst showed that the main transport mechanism in it is non-isothermal Knudsen diffusion. Another reason for this acceleration of DRM may be a change in the types of reactions involved in the main stages of the process.

Since this process is accompanied by a large endothermic effect and the appearance of a temperature difference in the parts of the reaction space adjacent to the membrane catalyst, it was to be expected that a temperature gradient would appear on the channel walls of the porous structure of MC. It is known [19] that the temperature gradient on the surface of narrow channels, for which the Knudsen number Kn ≥ 1, is the driving force that generates the phenomenon of thermal slip. In this case, it does not matter at all in which way a gradient is created on the surface of the channels formed in the pore structure. A necessary condition for the occurrence of thermal slip in a narrow channel is the fact that this channel connects two adjacent volumes of the reaction space, in which the gas has the properties of a continuous medium, and its thermodynamic characteristics, temperature, pressure, and concentration are different.

In work [18], the analogy between a membrane reactor and a Knudsen compressor is analyzed in detail. In this work, it is shown that the intensification of the DRM process under no isothermal conditions of Knudsen diffusion is primarily due to a change in the mechanism of mass transfer in the channels of the pore structure of the membrane catalyst, which can be accompanied by a change in the type of reaction involved in this process.

This paper presents the results of kinetic studies of DRM in a reactor with a membrane catalyst in which the active substance is molybdenum carbide (Mo_2_C). In a reactor with membrane catalyst, two contactor modes are analyzed, which can only be realized in this type of membrane reactor. In the kinetic experiment, in addition to the traditional parameters of the catalytic process, such process indicators as the pressure drop at the reactor inlet and the temperature of the reaction medium on both sides of the membrane catalyst were recorded. In the contactor mode with forced transport, the reaction mixture entered the reactor and was transported through the pore structure of the membrane catalyst due to excess pressure. In the contactor mode with diffusion transport, the excess pressure at the reactor inlet was negligible, and the components of the reaction mixture could penetrate into the pore structure solely due to diffusion.

Intensification of dry methane reforming on a membrane catalyst cannot be explained within the framework of existing views on the role of the membrane in the catalytic processes. The purpose of this study was to elucidate the reasons for the unusual intensification of this process by analyzing a kinetic experiment under DRM conditions in a reactor with a membrane catalyst in contactor modes with “forced” and “diffusion” transport of reagents, taking into account the phenomenon of thermal slip (thermal transpiration).

## 2. Materials and Methods

### 2.1. Membrane and Traditional Mo_2_C Catalyst Preparation

Membrane catalysts based on molybdenum carbide were obtained by CVD in a reactor with “cold” walls at atmospheric pressure, similar to that shown in [16,17]. On the outer surface of a two-layer corundum membrane with an outer selective layer of a mixture of molybdenum hexacarbonyl vapor with nitrogen, molybdenum dioxide was deposited in the form of a massive layer about 8 μm thick. The obtained sample, containing 4.6 wt% MoO_2_, with a length of about 7 cm was sealed on one side, and the open side was installed in the holder of the membrane reactor. The length of the working surface was approximately 5.5 cm. Then, using temperature-programmed carburization with a methane-hydrogen mixture, MoO_2_ was converted into Mo_2_C by heating to a temperature of 800 °C. After holding at this temperature in the carburization mixture, a mixture of reagents with a stoichiometric composition of CH_4_:CO_2_ = 1:1 was fed into the membrane reactor. Then, the temperature was raised to the maximum value of the kinetic experiment, and we began to study the modes of the contactor with diffusion and forced transport of reagents. This method is presented in more detail in the works [20,21].

Traditional powdered catalysts were prepared by the sol-gel method from dispersions of molybdenum blue, which were synthesized from ammonium heptamolybdate by reduction with ascorbic acid in acid medium (pH = 2.0). Ascorbic acid (molar ratio [R]/[Mo] = 1/1) and hydrochloric acid (molar ratio [H]/[Mo] = 0.8) were added to a solution of ammonium heptamolybdate (0.07 M) with vigorous stirring. The appearance of a bright blue color of the dispersion indicated the formation of molybdenum blue. The resulting product was subjected to thermal treatment of molybdenum blue xerogels in a flow of an inert gas (N_2_) at a final temperature of 900 °C [22].

Both catalysts, membrane and traditional (powder), had the same chemical and phase composition—β-Mo_2_C, η-MoC. The traditional (powdered) catalyst was of the bulk type. The phase composition of the catalysts—powdered and membrane—was established earlier according to X-ray analysis. A detailed analysis of these materials is given in the works [20,23].

In order to ensure the maximum similarity of both catalysts, the membrane catalyst was formed by placing the active component in the form of a massive layer on the outer surface of the cylindric membrane [19].

### 2.2. Kinetic Study of the Modes of a Membrane Contactor with Diffusion and Forced Transport of Gases

The kinetic study of traditional and membrane catalysts was carried out in different reactors in the same temperature ranges: 850, 870 and 900 °C—for powdered catalyst, 820, 850, 870 and 890 °C. The traditional catalyst was tested in a quartz reactor with a central thermocouple cover and a shelf for placing the test sample. The catalyst was mixed with crushed quartz, and the resulting mixture was loaded into the annular space of the reactor (between the thermocouple jacket and the reactor wall) onto a quartz layer. Coarse quartz was also poured on top of a layer of a mixture of catalyst and crushed quartz. The use of quartz in this experiment was due to the need to avoid “overcooling” of the catalyst under study.

The study in a membrane reactor was distinguished by the fact that at each temperature and each flow rate of the reaction mixture, the compositions of the reaction mixture at the outlet of the reactor were determined in two modes of the contactor sequentially (in the mode of “diffusion” and “forced” transport), changing the output the reaction mixture from the reactor. In the mode of “diffusion” transport, the reaction mixture left the reactor from the annular space, and in the “forced” mode, from the inner volume of the membrane catalyst. After each change in the regime, flow rate, and/or temperature, the reactor was maintained until all parameters were constant before determining the composition of the reaction mixture at the outlet of the reactor.

## 3. Results

### 3.1. Theoretical Basis of Gas Transportation in the Pore Structure of a Membrane Catalyst

#### 3.1.1. Transport of Gases in Porous Materials, Thermal Slip (Thermal Creep, Thermal Transpiration)

The transport of gases in porous materials in the modern theory of mass transfer is characterized by the Knudsen number (Kn), the numerical values of which determine the modes of gas flow in micro- and nanochannels of the pore structure of various materials. The Knudsen number is a dimensionless parameter determined by the ratio of the mean free path of a molecule (Λ) to the characteristic geometric diameter of the channel (L):(1)$$\mathrm{Kn}=\frac{\Lambda }{L}$$

When the value of the Knudsen number is greater than one, microscopic effects begin to manifest themselves more significantly. At the same time, such parameters as pressure drop (difference), shear stress, mass flow of matter and heat flow cannot be predicted (calculated) from models based on the continuity hypothesis.

At Kn < 0.01, the flows in the channels can be considered as obeying the laws of continuous media, and at Kn > 10, as micro flows obeying the laws of free-molecular flows of Knudsen diffusion. Sparse gases in the range of Kn numbers from 0.01 to 10 can exist simultaneously, both in the form of continuous and free-molecular flows. This interval is also divided into two regions [19]:

Kn = 0.01 ÷ 1—slip flow area;

Kn = 1 ÷ 10 transition area.

Later, during the experimental verification of this classification, it was found that, according to the dependence of the mass flow on the pressure difference in logarithmic coordinates, three modes can be distinguished. From Kn = 0.6 to 17, there is a transient regime in which the mass flow remains almost unchanged. This area is typical for relatively large channels and micro channels.

The temperature gradient directed along the axis of the channel in the pore structure causes an unusual (counterintuitive) phenomenon of mass transfer, called thermal creep, thermal transpiration, etc., which is characterized by the movement of molecules in the near-wall gas layer in the microchannel from the region with low temperature values to the region with high temperature values. As a result of this transfer of matter, the pressure in the volume of the working space with a low temperature decreases. In a heated volume isolated from the external environment, the pressure will increase. The resulting pressure difference in adjacent parts of the working space will cause a reverse flow of gases, which will move in the same channel along its longitudinal axis. These changes will occur as long as the flows do not compensate each other. Thus, a circulation loop will appear in the channel, which will include both of these flows and adjacent volumes of the reaction space. Thanks to such a circuit, intensive mass transfer will occur between adjacent volumes through the membrane catalyst (or membrane). The peculiarity of this contour is that the wall flow and the axial flow obey different laws. The wall flow is controlled by the laws of ideal gases, whereas the axial flow is controlled by the laws of continuous media.

Thermal slip of gases is initiated in the rarefied near-wall layer of gas of channels formed in the pore structure of the partition, dividing the working space of the device into two parts or connecting into two containers with a gas or a mixture of gases.

Numerous studies of the phenomena of rarefied gas transport in micro- and nanochannels that exist momentarily have been studied in depth; a theory and a mathematical apparatus have been created, which are presented in various monographs and give a fairly complete idea of the occurrence of the phenomenon of thermal slip and regularities characterizing it [20,24].

#### 3.1.2. Similarity of a Reactor with a Membrane Catalyst and a Knudsen Compressor

In the past few decades, many new devices have been designed that can be attributed to the fields of micro- and nanotechnology. These devices contain all modes of gas transport, characterized by the Knudsen number, the limiting value of which can reach 200. The characteristic geometric dimensions of these devices range from fractions of a nanometer to several microns. It is pertinent to note here that the characteristic dimensions of porous materials used for the preparation of heterogeneous catalysts also lie in this range. This means that in order to analyze the phenomena of mass transfer in heterogeneous catalysis and, especially, in membrane catalysis, it is necessary to take into account the peculiarities of micro- and nanophenomena.

One such device, which is based on the laws of micro- and nanoflows and whose characteristics lie in the range of Kn numbers coinciding with the range of heterogeneous catalysts, is the Knudsen microcompressor. Its design has no moving parts, and it is capable of pumping gases at virtually any vacuum.

The Knudsen compressor has been known since 1909 as an operating device without mechanically moving parts for pumping gases. The phenomenon on which the principle of its action is based was mathematically described and analyzed by J. Maxwell and O. Reynolds in the second half of the 19th century. However, the widespread use of these devices began at the end of the 20th century, when the need arose to create miniature devices for research in outer space. In works [25,26], a modern interpretation of the Knudsen compressor is given. The main elements of this device were analyzed, and the basic principles of their design were formulated, which made it possible to create both single-stage and multi-stage compressors for various purposes. In a Knudsen compressor, the temperature difference is created artificially by heating one container and cooling another. The temperature gradient can be created in any other way. For example, in work [25], temperatures were generated by passing an electric current through a porous partition made of semiconductor material. The temperature gradient can also be created in a porous diaphragm made of a catalytically active material or by applying such a material to the inner surface. When a chemical reaction with a noticeable thermal effect occurs in the channels of the pore structure of such a diaphragm, a temperature gradient will also be generated when the reaction mixture is passed through it. In work [19], the structural analogy of the membrane reactor and the Knudsen compressor was considered. It was experimentally shown that in the channels of the pore structure of the membrane catalyst, the mass transfer of gases is initiated by the temperature gradient, as in the porous diaphragm in the Knudsen compressor.

#### 3.1.3. Features of Non-Isothermal Knudsen Diffusion That Must Be Taken into Account in the Analysis of Kinetic Data and the Construction of a Kinetic Model of a Membrane Reactor

1. The necessary conditions for the occurrence of the phenomena of non-isothermal Knudsen diffusion (thermal creep, thermal transpiration) in porous diaphragms dividing the working space of various devices are: the state of gases in the pores (Kn number) and the presence of a temperature difference in volumes adjacent to the diaphragm.

2. The ordered movement of gases in the channels formed in the pore structure of the diaphragms separating the working space of various devices is due to the tangential temperature gradient along the length of the channels. The temperature gradient induces the simultaneous existence of conjugate gas flows in the diaphragm channels.

3. The flow of thermal slip in the channels of the pore structure is localized in the near-wall layer of the channel, and the movement of gases in this layer obeys the laws of rarefied media (ideal gases). Another flow arising in the same channel due to the pressure difference in the volumes adjacent to the channel is opposite to thermal slip and moves along the axis, obeys the laws of hydrodynamics of continuous media. The movement of both streams occurs without the use of mechanical devices that induce the transport of gases.

4. Under steady-state conditions (at a constant temperature gradient), these flows form a circulation loop that includes both volumes of the compressor working space adjacent to the diaphragm.

5. The flows that make up the circulation loop in a single channel are energetically conjugate. A change in the magnitude of one stream causes an equivalent change in the other. The maximum mass flow rate is determined by the channel permeability under the appropriate conditions (T, P, Kn).

6. The laws governing the dynamics of gases in the flows that make up the circulation loop in the channels of the pore structure make it possible to formulate an important conclusion for kinetic analysis: only molecules of the thermal slip flow interact with the surface of the channel walls in the porous structure of the diaphragm, whereas for molecules of the flow controlled by the difference in pressures, the walls of the channels for the reaction interaction are practically inaccessible. This means that homogeneous reactions in a thermal slip flow are impossible, since there are practically no intermolecular collisions in it. It can also be noted that the formation of a thermal slip flow implies the absence of a viscous boundary sublayer, which, in the theory of continuous media, is responsible for hydraulic losses in channels due to internal friction and roughness of the channel walls.

7. The transport of the reaction mixture in a rarefied thermal slip flow is characterized by the fact that, in the absence of intermolecular collisions, each component moves at its own average speed, inversely proportional to its molecular weight. It follows from this that the velocities of movement of the components of the reaction mixture in the flow of thermal slip should be redistributed according to the molar masses. In this case, the frequency of collisions with the channel surfaces by the molecules of the reaction mixture will change. It can be expected that the frequency of collisions with the catalytic surface of carbon dioxide molecules will be the lowest, and for hydrogen, the highest.

8. The simultaneous existence in the circulation loop of two gas streams, obeying different laws of dynamics, will obviously lead to the fact that in a reactor with a membrane catalyst the localization of heterogeneous and homogeneous reactions will occur in different volumes of the reaction space. Heterogeneous reactions should be localized in the near-wall flow of channels, and homogeneous reactions should be localized in flows obeying the laws of continuous media—in the annular space (shell side) of the reactor, in the inner volume of a cylindrical membrane catalyst (tube side), and in the volume of large pores (for example, in large pores of a membrane catalyst carrier). All of the above patterns should be taken into account when analyzing kinetic data. A detailed analysis of the flows is presented below.

#### 3.1.4. Non-Isothermal Gas Transfer and Flow Structure in a Membrane Catalyst Reactor in Contactor Modes with “Diffusion” and “Forced” Transfer

Earlier in [19], it was experimentally established that in the mode of a contactor with “diffusion” transport in a reactor with a membrane catalyst, under DRM conditions, the methane flux reaching (entering into the dissociation reaction of) the catalytic surface turned out to be an order of magnitude greater than the amount of methane, which diffuses through the pore structure of the membrane catalyst under isothermal conditions.

The contactor mode in a membrane catalyst reactor was studied in two versions. Figure 1 schematically shows a reactor with a membrane catalyst in the contactor mode with forced (FT) and diffusion (DT) transport of reagents (left sketch in Figure 1a,b) and the structure of flows in the cross section near the membrane catalyst (right sketch). The difference between the modes is only in the route along which the mixture of reagents is transported through its reaction volume. The mode in Figure 1a in publications on membrane catalysis is often called forced flow-through; we will call the regime a mode with forced transport of reagents, and the regime in Figure 1b, a mode with diffusion transport, in order to emphasize the formal difference in driving forces transporting reagents through the reactor.

The flows entering and leaving the reactor are caused by external forces (pressure drop at the inlet and outlet from the apparatus), and the flows circulating in the pores are caused by the temperature gradient and the pressure difference in the channels of the pore structure of the membrane catalyst. If in Figure 1d we exclude the flow of incoming reagents from consideration, then we will find a complete similarity of the flow structure in the reactor with a membrane catalyst and in the Knudsen compressor, for which the suction channel into the rarefied space of the device is closed (blue volume in the figure) [25,26]. The volume of the reaction space, colored in blue, corresponds to the reaction space inside the tubular membrane catalyst (tube side in MR), whereas the volume colored in red corresponds to the reaction volume of the annular space (shell side in MR). Under the conditions of the kinetic experiment presented below, under all studied conditions, a low temperature was recorded inside the membrane catalyst, and a high temperature was recorded in the red one. The resulting temperature difference between these volumes was a consequence of the endothermic effect of dry reforming of methane.

Streams (hereinafter ($Q_{T}^{F}$ *u* $Q_{T}^{D}$)), moving near the surface in the pore structure represent the thermal slip generated by the temperature gradient in the modes of forced and diffusion transport, respectively. The magnitude of the temperature gradient on the channel wall in the pore structure of the membrane catalyst will be determined by the total thermal effect of the reactions that take place on it. Streams $Q_{P}^{F}$ *u* $Q_{P}^{D}$, moving in opposite directions along the channel axis, are caused by a pressure difference (rarefaction) ($P_{h}\;-\;P_{c}$), which arises due to the molecules leaving from the closed internal reaction space in the membrane catalyst with flows $Q_{T}^{F}$ *u* $Q_{T}^{D}$ into the annular space (from tube side to shell side of membrane reactor). Streams ($Q_{T}$ *u* $Q_{P}$) are under stationary conditions of each mode, and circulation loops are formed in single channels (pores) connecting both parts of the reaction volume. In a kinetic experiment, the results of chemical interaction are always recorded under stationary conditions.

Thermal slip ($Q_{T}$) obeys the Boltzmann law (the law of molecular dynamics), and the flux ($Q_{P}$), in general, Darcy’s law (the law of dynamics of viscous continuous media). The driving forces of these flows cause the counter movement of these flows. These flows are energetically conjugated to each other, and an increase in one of them causes an increase in the other and vice versa. At the same time, it can be argued that the course of the heterogeneous catalytic stage on the membrane catalyst occurs only in the thermal slip flow, since it is the flow that contacts the catalytic surface.

In the diffusion mode of transport in a reactor with a membrane catalyst, the flow $Q_{T}^{D}$, which is a thermal slip, at constant temperature and pressure, will be almost constant. Its maximum value is determined by the average temperature of the pore surface and the temperature gradient on it, which ultimately depends on the activity of the catalyst and the heat effect of the reaction. $Q_{T}^{D}$ generates a transfer of molecules from the internal volume of the MC and creates a pressure difference (P_r_—P_x_), which is the driving force of mass transfer directed into the internal volume of the MK (stream $Q_{P}^{D}$). In this mode of transport in a reactor with a membrane catalyst, the flows $Q_{T}^{D}$ *u* $Q_{P}^{D}$ turn out to be equal quantitatively. In this case, the maximum values of the degree of conversion and selectivity of the process in the reactor will always be determined by the flow $Q_{T}^{D}$ at the appropriate process temperature and catalyst activity.

In contactor mode with forced transport, the excess pressure at the reactor inlet is added to the pressure difference generated by thermal slip (P_r_—P_x_), increasing flow $Q_{P}^{F}$. Accordingly, this increase will be accompanied by an increase in the thermal slip flux $Q_{T}^{П}$, since under stationary conditions the flows $Q_{P}^{F}$ *u* $Q_{T}^{F}$, are balanced. Obviously, in this mode it is possible to influence the flow $Q_{P}^{F}$, increasing it and compensating for the pressure loss in the pores, due to hydraulic resistance (internal friction and local resistance due to tortuosity and narrowing-expansion in the channels) until the maximum intensity of conversion on the catalytic surface is reached, corresponding in magnitude to a given temperature. With a large pressure drop at the reactor inlet in this mode, a decrease in the degree of conversion can be observed due to an increase in the average pressure in the pores and the removal of a part of the viscous flow with the product mixture flow leaving the reactor. In this case, an increase in the average pressure can adversely affect the thermal slip flux.

The above approach to the analysis of mass transfer for two modes of a contactor in a reactor with a membrane catalyst shows the key role of mass transfer processes in intensification by the example of DRM. In a kinetic experiment, the magnitude of these fluxes can be determined from the material and/or heat balances of the reactor. Considering that the reaction of carbon dioxide with hydrogen is not catalytic, it cannot proceed in a thermal slip flow, since there are no collisions between molecules in this flow [27].

The structure of micro-streams in a Knudsen compressor with a closed suction line absolutely coincides with the structure of streams in a reactor with a membrane catalyst operating in the mode of a DT-contactor. The inner space of the membrane catalyst in this mode of the contactor is isolated from the external environment, just like in the Knudsen compressor, in which the line of the gas entering it is closed.

In studies of the Knudsen compressor, this mode corresponds to the maximum static head (pressure mode) [26], and the flux $Q_{P}^{D}$ in this mode is equal to the thermal slip flux $Q_{T}^{D}$ at a constant temperature difference (constant gradient on the surface of the channels in the catalyst pores).

#### 3.1.5. Dry Reforming of Methane on Traditional Powder Catalysts

Dry reforming of methane (Reaction (1)) is a multistage process; however, there are still discrepancies in the interpretation of the mechanism by researchers. The most widespread phenomenological model of the DRM process today, substantiated experimentally [28], and which has been confirmed repeatedly in later publications [29,30], includes the following main provisions:(R1)$$CH_{4}+CO_{2}↔2CO+2H_{2}\;\Delta H_{298}^{0}=-247\;\mathrm{kJ}/\mathrm{mol}$$

The DRM process includes several sequential stages, in which three reactions occur:(R2)$$CH_{4}\to C_{s}+2H_{2}\;\Delta H_{298}^{0}=-75\;\mathrm{kJ}/\mathrm{mol}$$
(R3)$$C_{s}+CO_{2}\to 2CO\;\Delta H_{298}^{0}=-172.5\;\mathrm{kJ}/\mathrm{mol}$$
(R4)$$CO_{2}+H_{2}\to CO+H_{2}O\;\Delta H_{298}^{0}=-41\;\mathrm{kJ}/\mathrm{mol}$$

The authors of these works believe that the beginning of the process is the stage of dissociative adsorption of methane and carbon dioxide on the active sites of the catalyst. The decisive stage of CO_2_-reforming is the reaction of dissociation (cracking) of methane to hydrogen and carbon (R2), which proceeds with the participation of a catalyst. Both products of this reaction then participate in interactions with other components of the reaction medium. The carbon formed on the catalyst surface interacts with carbon dioxide (more precisely, with the dissociation product CO_2_—atomic oxygen) according to reaction (R3), which can be considered as the carbon dioxide gasification of active carbon. Another product of the dissociation of methane, hydrogen, also interacts with carbon dioxide, but already in the gas phase according to the reverse reaction (shift) of water gas (R4).

In reaction (R2), one of the three forms of carbon deposits observed in this process forms on the catalyst surface—“active” carbon [31]. Namely, active carbon interacts on the surface with oxygen formed during dissociative adsorption of CO_2_.

Unlike reaction (R2), reaction (R3) is “fast”. Reaction (R4) is also fast and almost always reaches equilibrium under DRM conditions, in which a byproduct, water vapor, is formed. Water vapor is formed with incomplete conversion of carbon dioxide (and methane, respectively).

The above three reactions (R2)–(R4) are sufficient to create a kinetic model of the DRM process in a traditional reactor with a fixed catalyst bed, which well describes the process at a stoichiometric ratio of reactants in the mixture fed to the reactor.

### 3.2. Kinetic Experiment and Analysis of Methane Carbon Dioxide Conversion Indicators in Conventional and Membrane Reactors

As can be seen in Figure 2a–d, the degrees of conversion of methane in a reactor with a membrane catalyst for the modes of the contactor in the investigated range of temperatures and flow rates, with the exception of the experiment at 820 °C (at which the experiment in a traditional reactor is not carried out), practically do not differ. However, the contact times required to obtain the same conversions in a membrane reactor in all experiments remain an order of magnitude less than in a conventional reactor.

In this case, the effect of excess pressure at the inlet to the reactor ($\Delta p=f\left(\tau_{\;cont}\right)$) (Figure 2a–d) on the degree of conversion of methane in both modes of the contactor in a reactor with a membrane catalyst in our experiment turns out to be insignificant or absent altogether. And this is observed under conditions when, in the mode of “diffusion” transport, the pressure difference at the inlet to the reactor is negligible, or rather, equal to the hydrodynamic resistance of the annular channel of the reactor, and in the mode of the contactor with “forced” transport, the excess pressure at the inlet to the reactor varies from 0.11 to 0.43 atm.

Researchers have paid attention to this feature of the membrane catalytic reactor for a long time [16]; however, this fact remains without a clear and convincing explanation. Thus, it can be assumed that the pressure difference in the membrane reactor cannot be considered a value that determines the transport of reagents to the inner surface of the catalyst, and the contact time in all modes and in the entire flow rate range is determined by a different driving force than in a traditional reactor.

In all experiments, the observed temperature differences $\left(T_{out}-T_{in}\right)$ on both sides of the membrane catalyst have positive values, although the effect of these values on the contact time visually differ depending on the temperature of the DRM process in a reactor with a membrane catalyst. That is, the temperature inside the membrane catalyst (i.e., in the inner space of the cylindrical membrane catalyst), which we have chosen as a characteristic of the DRM process, turns out to be lower than the temperature in the annular space of the membrane reactor over the entire studied range of values (from 820 °C to 890 °C). This means that a temperature gradient arises on the walls in the channels of the pore structure of the membrane catalyst under the conditions of the DRM process, directed from the shell side of the reactor to the tube side.

Table 1 shows the kinetic parameters of DRM on conventional and membrane catalysts. The rate constants (extrapolated at contact time = 0) of reaction (R2) in both modes of the membrane reactor are close at all temperatures and significantly exceed the constants on a traditional catalyst.

The activation energy (Table 1) in a traditional reactor is higher than in the membrane type, and its value corresponds to the values that we observed earlier [18,32] on such catalysts. The E_a_ values for the membrane catalyst, although different, are insignificant. In a reactor with forced transport of reagents, E_a_ has the smallest of all the values established in this experiment. The difference in the activation energy values in traditional and membrane reactors indicates a diffusion mechanism of transport in the membrane catalyst, which is quite consistent with the conditions of this experiment.

According to the values presented in (Table 1), it can be seen that the values of the rate constant on the membrane catalyst exceed the constants on the traditional catalyst. The ratio of the constants for both catalysts characterizes the intensification of reaction (R2) and the entire DRM process in the membrane reactor. From the data presented in Table 1, it can be seen that the rate constants in a membrane reactor increase with increasing temperature more slowly than in a traditional reactor. The ratio of the constants decreases with increasing temperature, but still, under the conditions of this experiment, it remains quite high. This means that the temperature dependence of the constants on the membrane catalyst differs from the exponential dependence on a traditional catalyst. Obviously, there are temperatures at which the rate constants become close and the membrane catalyst will lose its advantages over the traditional catalyst.

Small differences in the values of the rate constants in the membrane reactor give grounds to assume that in both modes of RMC, driving forces of similar magnitude and origin act, causing the activated transport of reagents through the pore structure of the membrane catalyst. At the same time, both rate constants on the membrane catalyst significantly exceed the constant for a traditional catalyst, and the difference between them in both modes is much less than the difference between these rate constants and the rate constant for a traditional catalyst. This means that a factor arises and acts in a membrane reactor, accelerating the catalytic process in it in comparison with the process in a traditional reactor. The reasons for the acceleration of the catalytic process, in the general case, can be changes in the chemical or mass transfer stages of the catalytic act.

The presence of a temperature difference in the reaction volumes adjacent to the membrane catalyst, very large values of the reaction rate constant and the characteristics of the pore structure of membrane catalysts indicate the presence of conditions necessary for the occurrence of thermal slip in the channels of the pore structure of the membrane catalyst.

The identity of chemical processes or their absence when comparing membrane and traditional catalysts can be judged from the dependence shown in Figure 3 and Figure 4. The trend lines in these figures illustrate the regularities of the consumption of reagents and the formation of reaction products. The trend lines in these figures represent the ratio of the rates of consumption or formation of each component of the reaction mixture to the rate of consumption of methane. The dependences of changes in the concentrations of CO_2_, CO, H_2_ and H_2_O in the modes of the contactor with diffusion (orange symbol) and forced (black symbol) transport confirm the complete identity of the chemical and transport stages in both modes of the membrane reactor. For all components of the reaction mixture in both modes of the membrane reactor, the concentrations lie on the same trend lines in Figure 3 and Figure 4.

However, the trend line for hydrogen plotted on the basis of the results obtained on the traditional catalyst with Mo_2_C indicates a difference between the processes of hydrogen formation on the traditional and membrane catalysts.

The difference between the processes on the traditional and membrane catalysts is especially noticeable in the dependences of the formation of water vapor in Figure 4. This figure shows that the formation of water on a traditional catalyst significantly depends on the temperature of the process. On a membrane catalyst, the experimental points at all temperatures and flow rates of the reaction mixture lie on the same trend line, whereas on a traditional catalyst, a unique trend line is observed for each temperature. Moreover, as the conversion of methane increases, they each approach the trend line for the membrane catalyst.

The formation of water vapor on a conventional catalyst, as discussed above, is due to the interaction of hydrogen with carbon dioxide via reaction (R4). This reaction is homogeneous and noncatalytic, and its mechanism cannot change depending on the type of catalyst. Since it is difficult to imagine why the reaction mechanism (R2) can change on the same catalyst, it can be assumed that the change in the trend line of water vapor is due to a change in the transport of reagents to the catalyst surface.

As noted above, a feature of transport on a membrane catalyst is the formation of a circulation loop, which occurs under stationary conditions of the DRM process. The main feature of the circulation loop is that the flows that participate in it obey different physical laws. As a result, the localization of the main stages of the process in different parts of the working volume changes in the membrane reactor. Reaction (R4) is localized only in those parts where the reaction mixture obeys the laws of continuous media, and the transport of all components to the catalytic surface is carried out by a thermal slip flow (in Figure 1 flows $Q_{T}^{F}$ and $Q_{T}^{D}$), which obeys the laws of rarefied media. Hydrogen molecules will be delivered to the catalyst surface at the highest speed. On the contrary, CO_2_ molecules will reach the surface at the lowest speed, in accordance with the reciprocal of their molar mass.

From this it follows that for water vapor, the probability of interaction with carbon deposits on the catalyst surface becomes higher than for carbon dioxide.
(R5)$$C_{s}+H_{2}O↔CO+H_{2}\;\Delta H_{298}^{0}=131.3\;\mathrm{kJ}/\mathrm{mol}$$

This reaction turns out to be more likely than reaction (R3), since the flow $Q_{T}$ is in a rarefied state, and the speed of water vapor moving in it is higher than the speed of CO_2_ molecules, which have a large molar mass.

Thus, as follows from the analysis of the kinetic experiment presented above, the phenomenological scheme of the DRM process on a membrane catalyst includes three main reactions: cracking of methane (R2), the reaction of hydrogen with carbon dioxide (II), and the interaction of water vapor with carbon deposits on the catalytic surface (III). In this case, some contribution to the result of the DRM process from the interaction of carbon dioxide with carbon deposits (R3) is possible.

On a membrane catalyst, water is an intermediate substance, and on a traditional catalyst, it is the final product. It should also be noted that two heterogeneous reactions occur on the surface of the membrane catalyst that are responsible for the temperature gradient in the channels of its pore structure—reactions (R2) and (R5). Namely, the total endo effect of heterogeneous reaction (R2) and (R5) determines the magnitude and distribution law of the temperature gradient in the channels of the pore structure.

#### Assessment of the Degree of Intensification of the Catalytic Process in a Membrane Reactor

The kinetic experiment presented above and its analysis suggest the following mechanism for intensifying the DRM process. This mechanism is based on the following facts:

1. In a reactor with a membrane catalyst, there are two independent driving forces of mass transfer. An external driving force is controlled by excess pressure at the inlet to the reactor and ensures the transport of reagents through the reactor.

2. Internal driving force provides intensive transport of the reaction mixture through the pore structure of the membrane catalyst. It arises because of the appearance of a temperature gradient on the walls of the channels in the porous structure of the membrane catalyst.

Mass transfer in the channels of the pore structure occurs in flows that form micro-circulation loops. In this case, there is an intensive mass transfer of the components of the reaction medium from the shell side of the reactor to the tube side of the membrane catalyst and back.

From the analysis of the transport of the components of the reaction mixture, it is not difficult to estimate the ratio of the residence time of the reagent mixture $\tau_{r}$ в membrane reactor and the time of its contact with the catalyst surface $\tau_{cont}$. If we follow the existing hypothesis about the limiting role of methane dissociation (reaction I) and use in the calculations the amount of methane entering the reactor and the amount of the reacted, which should be equal to the thermal slip flux $Q_{T}$, then the following relationships can be written:(2)$$\tau_{r}=V_{s}/Q_{CH_{4}}^{in}$$
(3)$$\tau_{cont}=\frac{\;V_{p}}{Q_{T}}=\frac{\delta_{Ct}F_{p}}{Q_{T}}$$
(4)$$N_{circ}=\frac{r}{\tau_{cont}}=\frac{k_{MC}}{k_{TC}}$$
where: $\tau_{r}$—residence time of methane in the reaction (annular) volume, s;$\;V_{s}$—reaction space volume, cm^3^; $Q_{CH_{4}}^{in}$—volumetric methane consumption, cm^3^/s; $\tau_{cont}$—contact time of the thermal slip flux with the surface of the active catalyst layer, s; $V_{p}=m\times V_{}$—the volume of (meso-) pores of the membrane catalyst, cm^3^; $\delta_{Ct}$—the depth of propagation of the active component of the catalyst in the pores of the membrane, cm; $F_{p}—$cross section of pores on the surface of a membrane catalyst, cm^2^; $N_{circ}$—circulation rate.

If we consider the conditions under which almost complete conversion of methane is achieved, then it can be assumed for the mode of a contactor with diffusion transport that all the incoming methane enters into a reaction, i.e., $Q_{CH_{4}}^{in}\;$=$\;Q_{T}=Q_{P}$). Then it is possible to estimate the limiting relationship between the residence time and the contact time. If we take into account only the volume of mesopores, the surface of which is predominantly involved in heterogeneous catalytic reactions and the volume of the annular space of the reactor, then in our experiments this indicator reached three orders of magnitude:$$N_{circ}\;=\;\frac{V_{s}}{V_{p}\;}\;=\;\frac{\pi \left(R^{2}-r^{2\;}\right)l}{m\times V_{\;}}=\;\frac{3.14\times \left(1.5^{2}-0.5^{2}\right)\times 5.0}{4.3\times 0.0046}=1590\sim 10^{3}$$
where: R—inner radius of the laboratory reactor, mm; r—outer radius of the membrane catalyst, mm; m—the mass of the membrane catalyst involved in the reaction, g; $V_{}$—specific pore volume (based on low temperature nitrogen adsorption), cm^3^/g.

The ratio of residence time and contact time in the first approximation can be considered as the multiplicity of the circulation of the reaction mixture in the pores of the membrane catalyst. With such a large circulation rate of the reaction, a large difference in the rate constants on the membrane and conventional catalysts becomes obvious. Moreover, if a reactor with a membrane catalyst is approximated by a cascade of ideally mixed reactors, then such a multiplicity is excessive, from the point of view of the problem of complete conversion of the reactants into products. This means that a reactor with a membrane catalyst has great potential for creating on this basis a compact, high performance and highly selective device for the process of carbon dioxide conversion of methane, by increasing the volume occupied by the membrane catalyst in the reaction space of the apparatus.

## 4. Conclusions and Outlook

Intensification of the catalytic process is the goal of almost any research in the field of catalysis, even if it is not directly stated. The main path followed by the overwhelming majority of researchers is to select or improve the chemical composition of the active component. However, not all seemingly promising catalyst compositions meet expectations. One of the reasons for such failures may be the limited transport of reagents to the catalyst surface (kinetic limitations). The membrane catalyst compares favorably with other types of heterogeneous catalysts. In a reactor with such a catalyst, it becomes possible to influence the mass transfer component of the catalytic process, for example, using excess pressure (or vacuum) to intensify the catalytic process. On membrane catalysts, the rate constant can be increased by orders of magnitude compared to a traditional catalyst with the same chemical composition of the active substance.

Under non-isothermal conditions in the channels of the pore structure, corresponding to the Knudsen diffusion conditions, a unique type of mass transfer arises, in which gas molecules move from an area of low temperature and low pressure to an area of high temperature and high pressure. These regions of the reaction volume are always formed when a membrane catalyst is placed in the reactor. The pore structure of a membrane catalyst can be considered as a set of channels connecting adjacent volumes of the reaction space. Thus, in a reactor with a membrane catalyst, it becomes possible to use adjacent volumes of the reaction space to control the transport of gases in the channels of its pore structure. To the traditional driving forces for membrane processes—the pressure difference and the concentration difference—the temperature difference should now be added. This will make it possible to put the phenomenon of thermal transpiration into practice and thereby expand the possibilities of mass transfer control in the processes of membrane separation and membrane catalysis.

In the pore system of the membrane catalyst, a new structure of mass transfer flows is formed, which causes the intensification of the chemical process. When stationary conditions are established in a reactor with a temperature gradient, thermal slip occurs on the walls of the channels of the pore structure, and the reverse flow compensates and creates a circulation loop. In this circuit, one flow obeys the laws of gas dynamics, and the other—the laws of classical hydrodynamics. Both flows that make up the circulation circuit in the channel are energetically conjugated; a change in one of them will cause a change in the other.

Comparing the residence time of the mixture in the annular space of the membrane reactor with the time of single contact of the reaction mixture with the catalytic layer of the membrane catalyst, it follows that this ratio can reach three orders of magnitude. This means that the actual contact time also increases by three orders of magnitude, due to this multiplicity of interaction of the reaction mixture with the catalyst surface.

The emergence of a circulation loop with conjugate flows in the pore channels of a membrane catalyst in multistage reactions can be accompanied by the localization of homogeneous and heterogeneous stages in different volumes (parts) of the reaction space. The localization of individual stages can result in a change in the mechanism of the process at different stages. For example, in the DRM process, instead of reacting carbon dioxide with carbon deposits on the catalytic surface, it is more likely that water vapor interacts with them. Another feature of the flow structure resulting from the process of thermal transpiration is the existence of mass transfer in the absence of a pressure difference at the inlet and outlet of the gas mixture from the apparatus. This feature manifested itself when comparing two modes of the contactor (with “diffusion” and “forced” transport of reagents). In the mode of the contactor with “diffusion” transport, the efficiency of the catalytic process remained as high as in the mode with “forced” transport, and hydraulic losses during the movement of the reaction mixture through the annular space of the reactor were flashing to zero.

It should be borne in mind that the intensification of mass transfer in the channels of the pore structure can exist only under conditions when the direction of the temperature gradient on the walls of the channels coincides with that of the pressure gradient. In the case of multidirectional action of temperature and pressure gradients, mass transfer in the channels of the pore structure can be significantly hampered or stopped altogether.

The thermal effect of this process is the sum of the thermal effects of reactions (1)–(3). However, the temperature gradient on the surface of the channels of the pore structure of the membrane catalyst will be determined by the sum of the thermal effects of only heterogeneous reactions (R2) and (R5).

If we consider the material presented in this article as the first attempt to explain the effect of the intensification of catalytic processes in a reactor with a membrane catalyst (described in [Alexandrov et al. [32], Gavrilova et al. [17,18]], a hypothesis based on a new approach to catalysis utilizing the phenomenon of thermal transpiration, it seems appropriate to us to note that it is difficult to consider all aspects of the proposed approach to the membrane reactor. The article considers only one of the many possible modes of a reactor with a membrane catalyst—the contactor mode. Analysis of other regimes remains a task for the future. A novelty due to the phenomenon of thermal transpiration is the occurrence of circulation circuits in the pore channels of the membrane catalyst, which provide a more intensive transport of reagents to the active surface of the catalyst. This hypothesis undoubtedly requires verification both in different catalytic processes and in different modes of the reactor with a membrane catalyst. However, even today it allows us to see the limits of its applicability. In particular, it is applicable to the conditions that determine the boundaries of the existence of thermal transpiration. From the foregoing, it can be assumed that the membrane catalyst reactor is preferably used at low pressures (close to atmospheric) or at reduced pressures. Moreover, such a reactor will be effective at almost any rarefaction. The achieved acceleration supports the expectation that such an approach will make it possible to create compact and small-sized reactors for reactions complicated by kinetic restrictions. We are confident that the forthcoming works and publications, both by our group and other researchers, will expand our understanding of the intensification of catalytic processes in reactors with a membrane catalyst.

Popular wisdom says that everything new is the forgotten old. Brilliant work a century ago by Martin Hans Christian Knudsen is finally beginning a new life in heterogeneous catalysis.

## Figures and Tables

**Figure 1 membranes-12-00136-f001:**
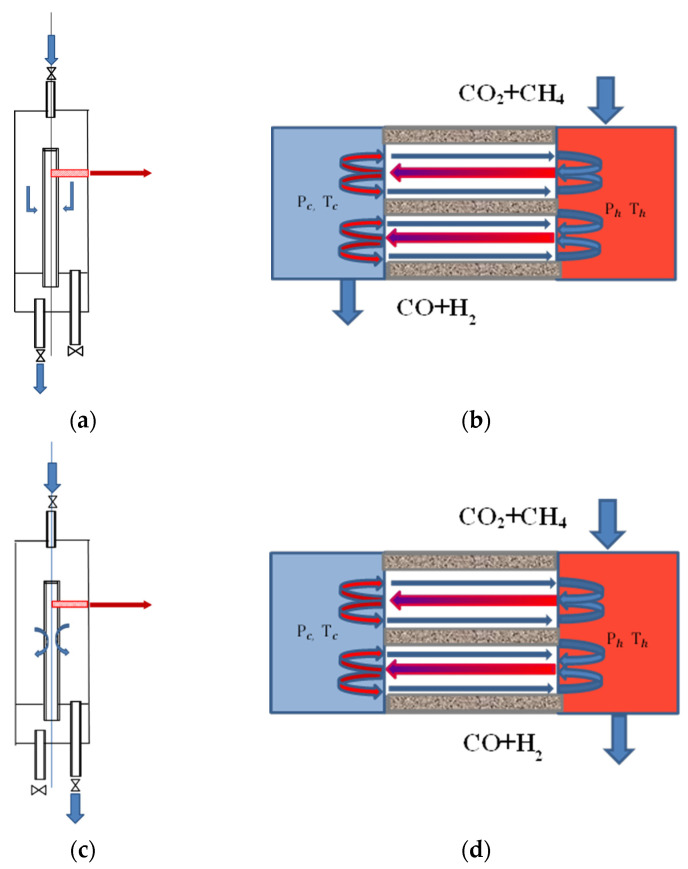
Illustration of the flow structure in the cross section of the membrane catalyst wall in the contactor modes with forced (**a**,**b**) and “diffusion” (**c**,**d**) transport. 
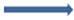
—thermal slip $Q_{T}^{F}$ (in FT-mode (**a**) and $Q_{T}^{D}$ (in DT-mode (**c**)); 
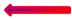
—viscous flow $Q_{P}^{F}$ (in FT-mode (**a**) and$\;Q_{P}^{D}$ (in DT-mode (**c**)); *P и T*—temperature and pressure in the internal (“cold”) and annular (“hot”) volumes of the reactor, respectively.

**Figure 2 membranes-12-00136-f002:**
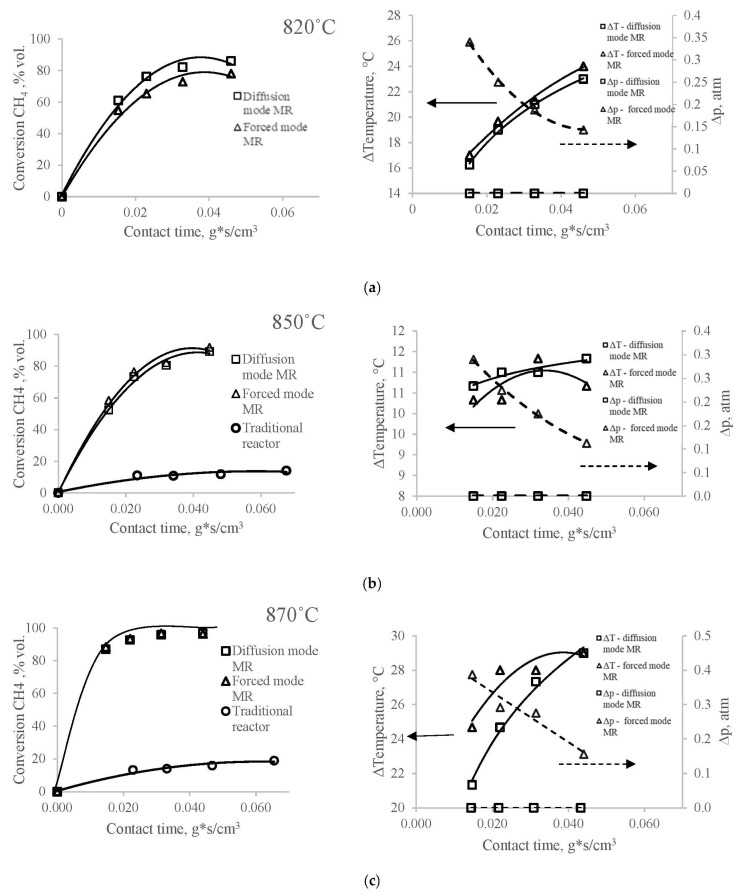
Dependence of methane conversion, temperature difference and pressure drop in reactors with membrane catalyst in contactor mode with forced and diffusion transport of reactants and with traditional catalyst on contact time at temperatures 820, 850, 870 и 890 °C (**a**–**d**, respectively). DRM: CH_4_:CO_2_ = 1:1. MC: weight of catalytically active component—0.153 g (Mo_2_C); TC: weight of catalytically active component—0.261 g (Mo_2_C).

**Figure 3 membranes-12-00136-f003:**
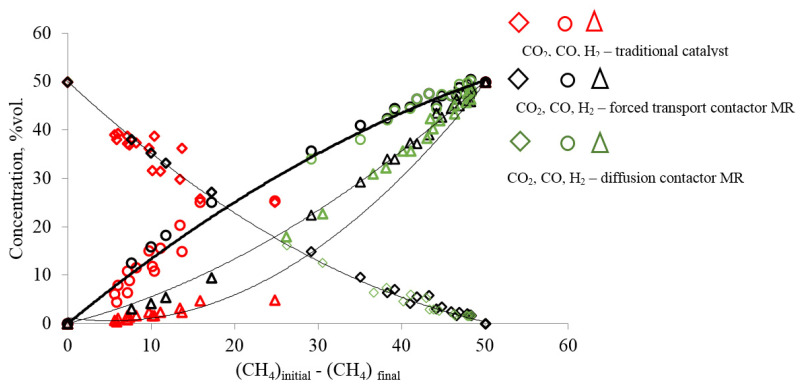
Dependence of the change in the concentrations of CO_2_, CO and H_2_ on the change in the amount of methane reacted on the membrane and traditional catalysts, when the temperature of the DRM process changes from 820 °C to 890 °C on the MC, and the flow rate of the reagent mixture changes from 40 to 150 cm^3^ (at standard)/min in the modes of the contactor with diffusion and forced transport of the reagent mixture. (For a membrane catalyst in a mode with forced transport at 820 °C, the interval of the consumption of the reagent mixture was increased and ranged from 40 to 320 cm^3^/min).

**Figure 4 membranes-12-00136-f004:**
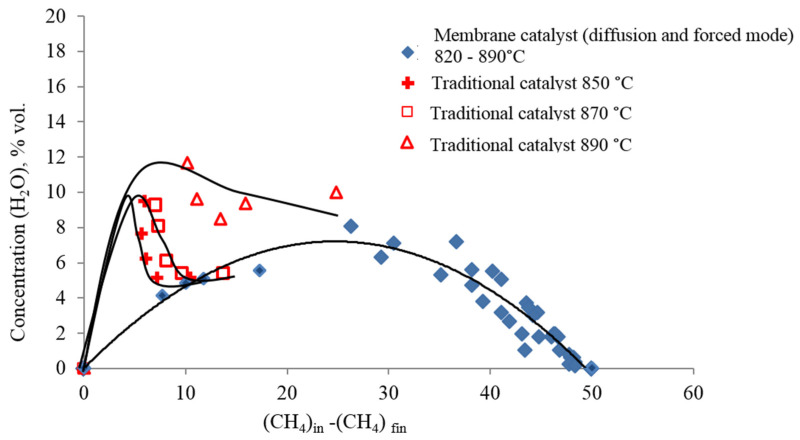
Dependence of the change in the concentration of H_2_O on the amount of methane that enters into the reaction on traditional and membrane catalysts, when the temperature of the DRM process changes from 850 °C to 900 °C at the TC and from 820 °C to 890 °C (at MC), and the flow rates of the reagent mixture change from 20 to 340 cm^3^ (at standard)/min and from 70 to 320 cm^3^ (at standard)/min, respectively).

**Table 1 membranes-12-00136-t001:** The value of apparent energy activation and specific rate constants of the reaction (I) in the modes of diffusion (DT) and forced (FT) transport with a constant in a conventional reactor.

Reactor Regime T °C	ki (cm^3^/g * s)			k_i_/kTC	
	TC	MC-DT	MC-FT	MC-DT/TC	MC-FT/TC
820	-	43	43	-	-
850	1.23	50	55	41	45
870	1.81	77	76	42	42
890	-	80	75	20 *	19 *
900	3.94	-	-	-	-
Ea, kJ/mol * K	315	103	92		

*—ratio of constants in a reactor with MC at T = 890 °C to a constant in a traditional reactor at T-900 °C, Legend in the table: TC—traditional catalyst, MC-DT—membrane catalyst (mode of “diffusion” transport), MC-FT—membrane catalyst (mode of “forced” transport), k_i_—reaction (R2) rate constant.

## Data Availability

The data presented in this study are available on request from the corresponding author.

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
