# Peer review of "Intensification of Dry Reforming of Methane on Membrane Catalyst: Confirmation and Development of the Hypothesis"

_membranes, 2022, doi:10.3390/membranes12020136_

Round 1
Reviewer 1 Report
This paper presents results related to methane dry reforming on porous membrane catalyst presented by team which actively works in this field and published a lot of papers on this subject in the International Journals, including Membranes 2021, 11, 497 . Qualification of this team is proper, methods are adequate, discussion is reasonable, conclusions are sound. So paper can be accepted for publication after making minor revision to bring it to standards on international publication. It seems that authors in a hurry made a lot of errors, typical examples (not all) are given below
- Page 14, line 509, “its value corresponds to the values that we observed earlier [19, 29]”. Reference [19] does not correspond to papers of this team.
- 3. : in the inset symbols corresponding to data for diffusion contractor MR are yellow, while in curves they are green.
- Text before Table 1 (page 14, line 499) “The rate constants (at contact time = 0)” is apparently puzzling from the physical point of view, since at zero contact time no reaction occurs. Proper formulation should be used such as “extrapolated to zero contact time”, etc, since efficient first order rate constants are usually estimated at small conversions when reagents concentrations are changed only slightly.
- 4 caption is clearly nonadequate
- Authors are advised to remove statement about reverse water gas shift reaction occurring only in the gas phase without any action of a catalyst. It is well known that traditional water gas shift catalysts (copper based, etc) catalyze both forward and reverse reactions, so at higher temperatures and longer contact times jst equilibrium composition is attached.
- Authors are also advised to clearly present data on hydrogen selectivity of membrane separation, since for porous membranes along with hydrogen other gases are present as well in permeate ( Ferreira-Aparicio et al. / Applied Catalysis A: General 237 (2002) 239–252)
- English is to be thoroughly checked and polished. Just one example showing typical misprints is that in page 17, line 644, sm3 is present instead of cm3, etc etc.
Author Response
Responses to the comments of the Reviewer 1
The authors are grateful to the Reviewer for careful consideration of the manuscript and comments.
Page 14, line 509, “its value corresponds to the values that we observed earlier [19, 29]”. Reference [19] does not correspond to papers of this team.
The authors agree with the remark, the error has been corrected
in the inset symbols corresponding to data for diffusion contractor MR are yellow, while in curves they are green.
The error has been corrected
Text before Table 1 (page 14, line 499) “The rate constants (at contact time = 0)” is apparently puzzling from the physical point of view, since at zero contact time no reaction occurs. Proper formulation should be used such as “extrapolated to zero contact time”, etc, since efficient first order rate constants are usually estimated at small conversions when reagents concentrations are changed only slightly.
The authors agree with the remark, сorrections made to the text of the manuscript
4 caption is clearly nonadequate
Normal figure caption restored
Authors are advised to remove statement about reverse water gas shift reaction occurring only in the gas phase without any action of a catalyst. It is well known that traditional water gas shift catalysts (copper based, etc) catalyze both forward and reverse reactions, so at higher temperatures and longer contact times jst equilibrium composition is attached.
The reviewer touched on one of the most controversial points in the dry reforming of methane reaction of water gas shift, concerning the scheme of intermediate reactions in this process. To prove the validity of our assumption, we quote from the work (Cesar AM Abreu, Douglas A. Santos, Jose A. Pacıfico, and Nelson M. Lima Filho. Kinetic Evaluation of Methane-Carbon Dioxide Reforming Process Based on the Reaction Steps // Ind. Eng. Chem. Res., Vol. 47, No. 14, 2008) "Methane cracking was a heterogeneous catalytic reaction, while the reverse water gas-shift reaction was a homogeneous one and the consumption of carbon by carbon dioxide was a heterogeneous noncatalytic reaction". In addition to the above, we could refer to our own data, which have not yet been published. We studied the interaction of CO2 and H2 in a quartz reactor filled with crushed quartz at temperatures of 920˚C and 950˚C. In this experiment, the conversion of CO2 and H2 was at 920˚C 23.5% and 27.5% (at τ = 1s).
Catalytically inert quartz acted as a heat accumulator in this process, reducing the temperature drop of the gas mixture in the reaction zone.
Authors are also advised to clearly present data on hydrogen selectivity of membrane separation, since for porous membranes along with hydrogen other gases are present as well in permeate (Ferreira-Aparicio et al. / Applied Catalysis A: General 237 (2002) 239–252).
The concept that is the basis for the organization of the catalytic process in [Ferreira-Aparicio et al.] differs significantly from the concept presented in this work. The catalytic function in the work cited above is provided by a traditional (powdered) catalyst placed in the internal space of the selective membrane. The main function of the selective membrane is to change the chemical equilibrium by removing products from the reaction zone. Formally, this corresponds to the Le Chatelier principle. In such a membrane reactor, the chemical process is combined with the process of separating the reaction mixture by removing one of the products.
However, the combination of these processes does not lead to the intensification of the chemical reaction, since a change in the thermodynamic characteristic - the equilibrium concentration, for example, of hydrogen - which is provided by different diffusion rates of the components in its porous structure, is accompanied by an increase in diffusion resistance. Diffusion resistance from this layer to the membrane and the intrinsic diffusion resistance of the membrane are added to the diffusion resistance of gases in a traditional catalyst layer. This generally reduces the rate of supply of reagents to the catalytic surface and reduces the efficiency of the catalyst used.
The concept presented in this paper is based on a new physical phenomenon for catalysis - thermal transpiration (thermal slip). The necessary conditions for the occurrence of thermal slip are: a) the placement of the catalyst on the membrane and b) the separation of the reaction space by the membrane catalyst into two parts. Due to the thermal effect of chemical reactions, a temperature gradient appears on the surface of the channels of the pore structure, which intensifies mass transfer through the pore structure of the membrane catalyst and, in turn, increases the rate of chemical reactions, delivering more reagents to the active surface.
English is to be thoroughly checked and polished. Just one example showing typical misprints is that in page 17, line 644, sm3is present instead of cm3, etc etc.
Correction of English and typos carried out
Reviewer 2 Report
The study entitled “Intensification of dry reforming of methane on membrane catalyst: confirmation and development of the hypothesis” is based on the kinetic studies of dry methane reforming (DRM) in a reactor with a membrane catalyst (RMC) in the modes of a contactor with "diffusion" and "forced" mass transfer.
The our all impression of this manuscript is good.
However, I would like to suggest some minor corrections before accepting this manuscript.
- The authors are being encourage to highlight the other possible alternative strategies in the event if multidirectional action of temperature and pressure gradients occur where the mass transfer in the channels of the pore structures hampered.
- The authors have mentioned that membrane catalyst is best alternative to replace the traditional catalysts. As far our knowledge membrane catalysts are not compatible everywhere as many catalysts are reaction-specific. They are not effective everywhere because of the possible undesirable interaction with the reaction mixture and many other issues. So it will be appreciated if the authors mention few short coming associated with the membrane catalyst.
- In materials and methods section in line 164 and 165, the quantity of reactants should be mention.
- At Line-169 and 170, the authors are encouraged to explain why these three materials have same chemical and phase composition?
- In materials and methods section: The materials prepared in this work has to be characterized through any tool or technique to ensure the formation. The authors are encouraged to explain how they confirmed the formation.
- In Figure-4, X-axis description makes no sense
- The authors are being encouraged to add a short paragraph discussing the novelty of this work and a comparison of this work with the already conducted work by Gavrilova et al [17, 18], Mark et al [25], Usman et al [26] and Alexandrov et al [29] mentioned in this manuscript.
- Valid references are required to be quoted at certain places, such as “sol-gel method” at line-164 (Page-4) and at many places between line-280 to 330.
- Certain references such as, 18, 19, 23, 25, 27 and so on are not in the proper format of this journal (Titles are in upper hand letters). All the references need to be revisit for the possible corrections.
- Some typographic mistakes have been observed at many places throughout this manuscript such as in the introduction part, Line-32, the word “al-ways” needs to be replace by the correct word “always” similarly in Line-41, the word “tradition-al” needs to be replace by the word “traditional and same in line-133 and a word methane at line-95 (Page-3) and so on.
I would like to recommend this manuscript for publication in this journal after incorporating the above mentioned minor corrections.
Author Response
Responses to the comments of the Reviewer 2
The authors are grateful to the Reviewer for careful consideration of the manuscript and comments.
The authors are being encourage to highlight the other possible alternative strategies in the event if multidirectional action of temperature and pressure gradients occur where the mass transfer in the channels of the pore structures hampered.
The reviewer suggested considering alternative strategies in a membrane catalyst reactor under conditions of non-isothermal Knudsen diffusion (i.e., under conditions of a temperature gradient on the surface of pore channels in a membrane catalyst). In a reactor with a membrane catalyst, in our opinion (if we understood the reviewer correctly), one should consider the fundamental modes of operation of such a membrane reactor as such strategies. In addition to the contactor mode, in which the reagents are fed into the reactor in a pre-mixed state, the distributor and extractor modes can be distinguished. A prerequisite for the analysis presented below will be the fact that only the membrane catalyst is placed in the reactor. The entire catalytically active component is placed on an inert membrane, while the traditional catalyst (in the form of crushed or molded elements) is absent, although today such a concept (strategy) is considered more promising.
As shown in the article under review, several ways of organizing the DRM process are possible in the contactor mode, depending on which part of the reactor the mixture of reagents is introduced into (sell side or tube side) and from which part the mixture of reaction products is removed (shell side and tube side). When the mixture is supplied and the reaction products are removed in one zone of the reaction space, for example, using the shell side, the intensity of the process decreases, but not too much, while the hydraulic resistance drops sharply. When a mixture of reagents is supplied and excess pressure is used from the side of the shell side and the reaction mixture is removed from the side of the tube side in the presented experiment, it turns out to be positive. However, a significant increase in excess pressure at the reactor inlet can lead to a decrease in the intensity of the process due to an increase in the average pressure in the pores, which will adversely affect the mass transfer in the thermal sliding flow. Following the modern theory of gas transport under conditions of non-isothermal Knudsen diffusion, an increase in the average pressure in the pore channels will be accompanied by a decrease in the thermal slip mass flow. A similar dependence should also manifest itself in other modes of the membrane reactor. It should be noted that the extractor mode (the reagents are supplied as a mixture, and the products are removed in the form of two streams of permeate and retentate) is an intermediate or transitional mode from the mode with "forced" transport to the mode with "diffusion". From the foregoing, it can be assumed that the membrane catalyst reactor is preferably used at low pressures (close to atmospheric) or at reduced pressures.
The authors have mentioned that membrane catalyst is best alternative to replace the traditional catalysts. As far our knowledge membrane catalysts are not compatible everywhere as many catalysts are reaction-specific. They are not effective everywhere because of the possible undesirable interaction with the reaction mixture and many other issues. So it will be appreciated if the authors mention few short coming associated with the membrane catalyst.
We certainly share this point of view of the reviewer. From the explanations to the previous remark of the reviewer, one of the "disadvantages" of the reactor with a membrane catalyst can be formulated. Such membrane reactors will be inefficient at high pressures. Other, possibly more serious disadvantages of the membrane catalyst reactor are: a) the problem of compact arrangement of membrane catalysts in the form of a bundle and b) the achievement of tightness of the installation of the membrane catalyst in a metal case.
In materials and methods section in line 164 and 165, the quantity of reactants should be mention.
Additional information has been added to the procedure for the synthesis of molybdenum blue
At Line-169 and 170, the authors are encouraged to explain why these three materials have same chemical and phase composition?
Additional information has been added.
In materials and methods section: The materials prepared in this work has to be characterized through any tool or technique to ensure the formation. The authors are encouraged to explain how they confirmed the formation.
Detailed studies of the structure and texture of the catalysts used in this work have been studied in detail using various research methods in our previous works. Since this work is devoted to a different problem, the authors would not like to repeat information about the studies of the properties of catalysts in this section. References to the mentioned works are added to the text of the manuscript.
In Figure-4, X-axis description makes no sense
Normal figure caption restored
The authors are being encouraged to add a short paragraph discussing the novelty of this work and a comparison of this work with the already conducted work by Gavrilova et al [17, 18], Mark et al [25], Usman et al [26] and Alexandrov et al [29] mentioned in this manuscript.
If we consider the material presented in this article as the first attempt to explain the effect of the intensification of catalytic processes in a reactor with a membrane catalyst (described in [Aleksandrov et al. [29], Gavrilova et al. [17, 18]), a hypothesis based on a new for the catalysis of the phenomenon of thermal transpiration, it seems appropriate to us to note that it is difficult to consider all aspects of the proposed approach to the membrane reactor. The article considers only one of the many possible modes of a reactor with a membrane catalyst - the contactor mode. Analysis of other regimes is yet to be done in the future. A novelty due to the phenomenon of thermal transpiration is the occurrence of circulation circuits in the pore channels of the membrane catalyst, which provide a more intensive transport of reagents to the active surface of the catalyst. This hypothesis undoubtedly requires verification both in different catalytic processes and in different modes of the reactor with a membrane catalyst. However, even today it allows us to see the limits of its applicability. In particular, it is applicable to the conditions that determine the boundaries of the existence of thermal transpiration. From the foregoing, it can be assumed that the membrane catalyst reactor is preferably used at low pressures (close to atmospheric) or at reduced pressures. Moreover, such a reactor will be effective at almost any rarefaction. The achieved acceleration makes it possible to expect that such an approach will make it possible to create compact and small-sized reactors for reactions complicated by kinetic restrictions. We are confident that the forthcoming works and publications, both of our group and other researchers, will expand our understanding of the intensification of catalytic processes in reactors with a membrane catalyst. Therefore, we end our article with ironic wisdom, which has numerous confirmations.
Valid references are required to be quoted at certain places, such as “sol-gel method” at line-164 (Page-4) and at many places between line-280 to 330.
Corrected
Certain references such as, 18, 19, 23, 25, 27 and so on are not in the proper format of this journal (Titles are in upper hand letters). All the references need to be revisit for the possible corrections.
Corrected
Some typographic mistakes have been observed at many places throughout this manuscript such as in the introduction part, Line-32, the word “al-ways” needs to be replace by the correct word “always” similarly in Line-41, the word “tradition-al” needs to be replace by the word “traditional and same in line-133 and a word methane at line-95 (Page-3) and so on.
Mistakes corrected